# GEOMETRY IMAGE DIFFUSION: FAST AND DATA-EFFICIENT TEXT-TO-3D WITH IMAGE-BASED SURFACE REPRESENTATION

**Slava Elizarov, Ciara Rowles, Simon Donné**
Unity Technologies

## ABSTRACT

Generating high-quality 3D objects from textual descriptions remains a challenging problem due to high computational costs, the scarcity of 3D data, and the complexity of 3D representations. We introduce **Geometry Image Diffusion** (GIMDiffusion), a novel Text-to-3D model that utilizes geometry images to efficiently represent 3D shapes using 2D images, thereby avoiding the need for complex 3D-aware architectures. By integrating a Collaborative Control mechanism, we exploit the rich 2D priors of existing Text-to-Image models, such as Stable Diffusion, to achieve strong generalization despite limited 3D training data. This allows us to use only high-quality training data while retaining compatibility with guidance techniques such as IPAdapter. GIMDiffusion enables the generation of 3D assets at speeds comparable to current Text-to-Image models, without being restricted to manifold meshes during either training or inference. We simultaneously generate a UV unwrapping for the objects, consisting of semantically meaningful parts as well as internal structures, enhancing both usability and versatility.

## 1 INTRODUCTION

Automatic 3D object generation offers significant benefits across video game production, cinema, manufacturing, and architecture. Despite notable progress in this area, particularly with Text-to-3D diffusion models (Boss et al., 2024; Siddiqui et al., 2024; Wang et al., 2023), generating high-quality 3D objects remains a challenging task due to computational costs, data scarcity, and the complexity of typical 3D representations. For one, it is crucial that the generated objects can be re-lit within graphics pipelines, necessitating the use of physically-based rendering (PBR) materials, for which little data at scale exists. Furthermore, graphics pipelines predominantly use meshes as their primary 3D representation: processing these at scale is notoriously difficult due to their irregular graph structure. Most techniques instead generate an intermediate representation, which increases the burden of training data pre-processing and generated object post-processing.

We propose diffusing joint albedo textures and *geometry images* (Gu et al., 2002), a 2D representation of 3D surfaces, using a Collaborative Control scheme (Vainer et al., 2024). This enables 3D object generation from text prompts, as shown in fig. 1. The image-based representation allows us to repurpose existing image-based architectures, while the Collaborative Control scheme enables us to leverage pre-trained Text-to-Image models, considerably reducing the required training data and costs. Geometry images, and more specifically multi-chart geometry images (Sander et al., 2003), offer two great advantages over other shape representations: they impose no constraints on the topology of the generated object and inherently partition it into semantically meaningful parts, allowing for easier manipulation and editing.

We believe that GIMDiffusion opens up a promising new research direction in Text-to-3D generation, providing a practical and efficient approach that can inspire future advancements in the field. In summary, the advantages of GIMDiffusion include:

- **Image-based**: By leveraging existing 2D image-based models instead of developing new 3D architectures, we simplify both model design and training.
- **Fast Generation**: We generate well-defined 3D meshes in under 10 seconds per object, which could be further enhanced using distillation techniques.

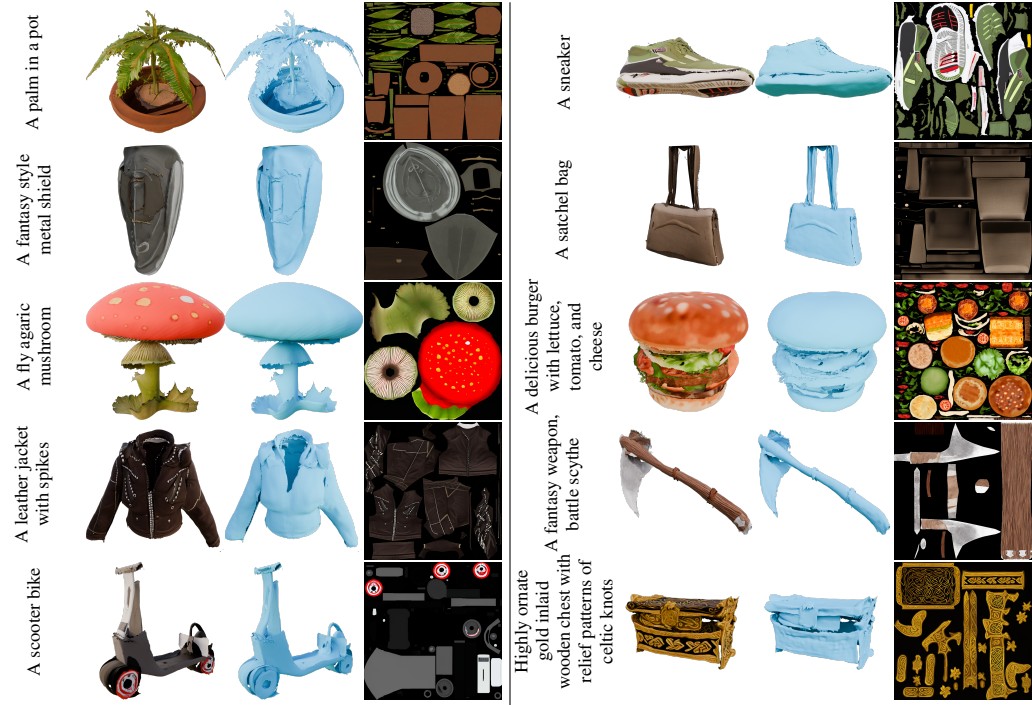

Figure 1: The objects are generated entirely using our method, including the structure, texture, and UV map layout, all created completely from scratch. For each object, we show the generated albedo texture, the textured mesh, the untextured mesh, and the respective text prompt.

- **Generalization**: Through collaborative control, we reuse pre-trained Text-to-Image priors, allowing for strong generalization beyond our limited 3D training data.
- **Separable Parts**: GIMDiffusion creates assets that consist of distinct, semantically-meaningful, separable parts, facilitating easier manipulation and editing.
- **Albedo Textures**: The 3D assets generated by GIMDiffusion do not have baked-in lighting effects, making them directly suitable for graphics pipelines.
- **UV unwrapping**: Our method jointly generates a UV unwrapping for the objects, learned from the training dataset. To the best of our knowledge, this is the first Text-to-3D method capable of achieving this.
- **Direct geometry**: Our 3D assets do not require the application of iso-surface extraction algorithms, which reduces potential artifacts and simplifies the overall workflow.

## 2 RELATED WORK

### 2.1 TEXT-TO-IMAGE GENERATION

Diffusion models (Sohl-Dickstein et al., 2015; Song et al., 2020) and flow matching (Lipman et al., 2022), alongside the rise of versatile, general-purpose architectures such as transformers (Vaswani et al., 2017), have brought considerable progress in generative modeling. In particular, text-conditioned image generation was revolutionized by approaches based on Latent Diffusion (Rombach et al., 2021) and its further extensions (Podell et al., 2023; Pernias et al., 2024; Esser et al., 2024). Foundational models like Stable Diffusion, trained on extensive internet-scale datasets (such as LAION-5B (Schuhmann et al., 2022)), are capable of generating complex scenes from text prompts while exhibiting an implicit understanding of scene geometry. Due to the high cost of training such models, they are often repurposed for other tasks or modalities (Zhang et al., 2023; Hu et al., 2023; Ke et al., 2023). Our proposed GIMDiffusion is a prime example of this, as it adapts the base model to specifically output albedo textures.

## 2.2 CONDITIONING DIFFUSION MODELS

Control mechanisms modify pre-trained foundational models, enabling them to accept additional conditions. Existing pixel-aligned control techniques fall into two categories: fine-tuning the base model with modified input and output spaces (Duan et al., 2023; Ke et al., 2023), or a separate model that alters the base model's internal states (Zhang et al., 2023; Zavadski et al., 2023). The latter approaches, such as ControlNet (Zhang et al., 2023), have gained wide adoption due to their ability to preserve the original model's performance while adding conditions such as human poses or depth images. AnimateAnyone (Hu et al., 2023) leverages a similar architecture to inject the base model's hidden states into a new branch that aligns with the base model's output.

In our method, we need to both control the base model (which will output UV-space albedo textures) and extract significant features from it (to generate the geometry image modality). Collaborative Control (Vainer et al., 2024) achieves exactly this by introducing bidirectional communication between both models, originally designed for Text-to-PBR-Texture generation.

## 2.3 TEXT-TO-3D GENERATION

We identify two main approaches to Text-to-3D generation: optimization-based and feed-forward methods. *Optimization-based* methods adapt 2D diffusion models to 3D by applying score distillation sampling (SDS) (Poole et al., 2022; Wang et al., 2022; 2023) to iteratively optimize a 3D scene, represented e.g. by NeRF (Mildenhall et al., 2020) or Gaussian Splats (Kerbl et al., 2023). These methods can produce content of high perceptual quality, but at the cost of impractically long generation times (Lorraine et al., 2023; Xie et al., 2024). The key advantage of this approach is its ability to utilize the rich 2D prior, allowing for 3D object generation without the need for expensive 3D data.

However, the lack of camera conditioning leads to discrepancies among different viewing angles (Janus effect (Poole et al., 2022)) and projection artifacts. To mitigate these issues, 3D-aware architectures and retraining on restrictive 3D datasets are often used, which can weaken the 2D prior (Shi et al., 2023b; Liu et al., 2023; Höllein et al., 2024; Zheng & Vedaldi, 2023; Shi et al., 2023a; Kant et al., 2024). Additionally, the original SDS formulation can lead to issues such as saturated colors, oversmooth geometry, and limited diversity (Wang et al., 2023; Zhu et al., 2023; Katzir et al., 2023; Alldieck et al., 2024; Liang et al., 2023; Wu et al., 2024).

*Feed-forward* methods directly generate 3D shapes without the need for iterative refinement. These methods employ complex, specialized architectures and typically require training on expensive 3D data from scratch. While seminal works like Point-E (Nichol et al., 2022) and its follow-ups (Huang et al., 2024; Zeng et al., 2022) demonstrate impressive generalization and diversity, the inherent lack of connectivity information limits the expressiveness of point clouds. Instead, many current methods rely on other proxy representations, such as neural implicits (Xie et al., 2021; Mildenhall et al., 2020; Malladi et al., 1995; Jun & Nichol, 2023; Zheng et al., 2022; Chen & Zhang, 2018; Mescheder et al., 2018; Yariv et al., 2023) or triplanes (Hong et al., 2023; Chan et al., 2021; Tochilkin et al., 2024; Boss et al., 2024; Bensadoun et al., 2024), to represent the objects.

Both groups of methods require pre- and post-processing to transform between the mesh domain and the proxy representation, e.g. through marching cubes (Lorensen & Cline, 1987) or tetrahedra (Doi & Koide, 1991). This process is costly and lossy, introducing issues such as quantization or grid-like artifacts, and leading to information loss, the loss of part segmentation and internal structures. Moreover, generated objects often require UV unwrapping, a complex post-processing step, for use in artist workflows. In contrast, our model jointly generates the UV map along with the object shape, requiring only a simple triangulation step.

## 2.4 GEOMETRY IMAGES

Geometry images (GIMs) (Gu et al., 2002; Sander et al., 2003) have been largely overlooked in deep learning (Sinha et al., 2016). XDGAN (Alhaija et al., 2022) is a pioneering effort that utilizes GIMs as the representation of choice for a StyleGAN-based architecture (Karras et al., 2018). However, due to the architectural constraints, the training data must be perfectly aligned with a template atlas, which limits its applicability to real-world data. Furthermore, the pre-processing algorithm provided in the paper is restricted to shapes of genus zero — Section 3.3.2 shows how to handle arbitrary shapes.

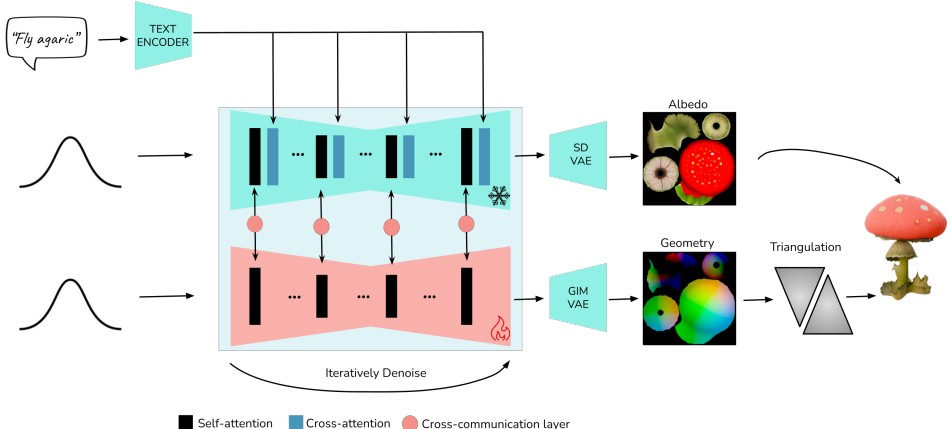

Figure 2: The method overview. Two separate diffusion models generate respectively albedo textures and geometry images. The former is a frozen pre-trained model, while the latter is an architectural clone trained from scratch. Models interact via the Collaborative Control Scheme. To obtain a mesh, generated geometry image is triangulated with the algorithm described in section 3.3.3

Recent concurrent work (Yan et al., 2024) has advocated for low-resolution $64 \times 64$ geometry images as a 3D representation for class-conditioned diffusion models and highlights its efficiency on a small-scale dataset of $8000$ objects (Collins et al., 2021), albeit with limited generalization. In contrast, GIMDiffusion addresses general Text-to-3D: rather than training a model from scratch, we leverage a pre-trained Text-to-Image diffusion model (using a Collaborative Control scheme trained on Objaverse (Deitke et al., 2022)) to retain generalization and diversity in the shapes, their appearance, *and* the UV atlas layout.

## 3 METHOD

### 3.1 GEOMETRY IMAGES

Geometry images (Gu et al., 2002) encode 3D surfaces in image format. This is achieved by a mapping function $\phi : [0, 1]^2 \to S \subset \mathbb{R}^3$, where $S$ denotes the 3D surface, and $[0, 1]^2$ represents the UV coordinates in the unit square, typically sampled on a uniform grid of the desired resolution. The function $\phi$, also known as a *surface parametrization* (Floater & Hormann, 2005), The choice of $\phi$ is crucial and is usually designed to minimize spatial distortion (e.g. conformal mapping).

Unlike 3D meshes, which require explicit data structures to maintain connectivity, geometry images implicitly connect each pixel to its neighbors. Gu et al. (2002) tackled surface mapping by cutting the input surface and warping it onto a disc, but this approach is limited to manifold objects and introduces significant distortions for high-genus shapes. Multi-Chart Geometry Images (Sander et al., 2003) address this limitation by mapping surfaces piecewise onto multiple charts of arbitrary shape, each homeomorphic to a disc. This approach adds flexibility (removing the manifold constraint) and reduces distortion (providing greater geometric fidelity). However, Multi-Chart Geometry Images lack connectivity information between charts and are susceptible to sampling artifacts, which can lead to visible cracks along boundaries, as discussed in section 4.5. Additionally, the algorithm proposed to construct $\phi$ is restricted to well-behaved manifold meshes. The optimal surface parametrization remains an active area of research (Sawhney & Crane, 2017; Srinivasan et al., 2023).

Most available 3D meshes have textures applied through UV maps. We observe that these handcrafted maps can be used to construct a desirable multi-chart $\phi$. Beyond providing low-distortion mappings, these UV maps strategically place chart boundaries in locations optimal for animation and downstream processing tasks. Furthermore, charts in handcrafted UV maps often *carry semantic meanings*, which propagate to the output of our model. This is evident in fig. 3, where the hands, face, and various parts of the gunslinger's appearance are separated in the UV atlas.

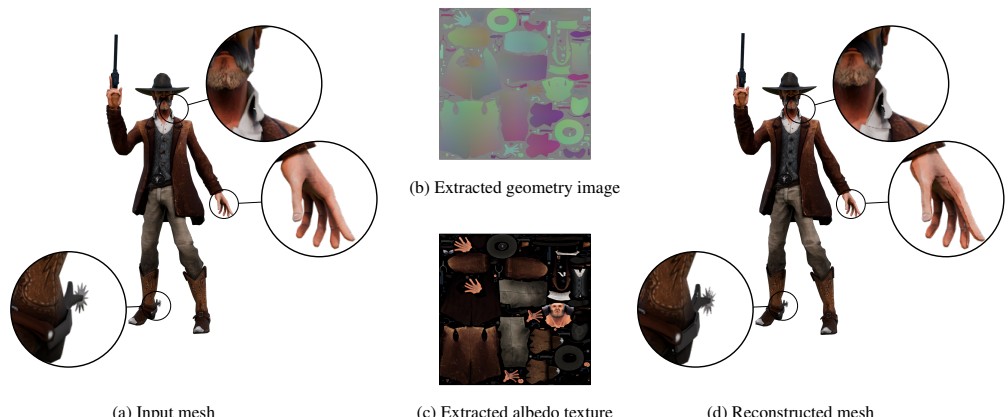

(a) Input mesh        (b) Extracted geometry image        (c) Extracted albedo texture        (d) Reconstructed mesh

Figure 3: (a) Ground-truth geometry, (b) geometry image and (c) albedo texture from our data pre-processing, and (d) the reconstruction using our dedicated VAE. We note the highly separable nature of the ground truth object, which is split into small components. The only visible artifact after decoding is the missing connection between charts of the geometry image, as discussed in sections 3.3 and 4.5.

As the density of the generated geometry's triangulation is limited by the resolution of the underlying geometry image, we follow Rombach et al. (2021) and use a VAE to increase the effective resolution of our model. To address the irregularities in geometry images and better match their distribution — particularly the need to accurately reconstruct the discontinuities at the boundaries of the charts — we add a channel to represent the multi-chart mask and modify the loss function accordingly. Otherwise, we follow the VAE training procedure from StableDiffusion1.5 (Rombach et al., 2021), leaving out only the GAN and LPIPS losses (see details in appendix A). As shown in fig. 3, the reconstruction using this VAE reconstructs everything well, except for the missing connections between the charts.

## 3.2 COLLABORATIVE CONTROL

To leverage the prior knowledge encoded in existing 2D Text-to-Image models, we use the Collaborative Control approach (Vainer et al., 2024). As shown in fig. 2, this approach comprises two parallel networks: a pre-trained RGB model and a new model for the geometry image. The former is responsible for generating UV-space albedo textures, while the latter generates the geometry images. These two models are connected by a simple linear cross-network communication layer, which allows them to share information and collaborate in generating pixel-aligned outputs across these different modalities. Crucially, this also enables the geometry model to influence the frozen model, guiding it to generate UV-space textures that would otherwise lie at the fringes of its training distribution. The frozen base model also drastically reduces the amount of data required to train the joint model while retaining generalizability, diversity, and quality (Vainer et al., 2024).

## 3.3 DATA HANDLING

### 3.3.1 DATASET

We train our model on the Objaverse dataset (Deitke et al., 2022). We curate this dataset to include only objects with both high-quality structures and semantically meaningful UV maps by filtering out 3D scans and low-poly models. The final dataset contains approximately 100,000 objects. Each data entry is accompanied by captions provided by Cap3D (Luo et al., 2023) and Hong et al. (2024). During training, we randomly sample these captions and apply random rotations of 90, 180, or 270 degrees to the extracted texture atlases. We now discuss how to transform these meshes into geometry images and back: the entire pre-processing was performed on consumer-grade PC hardware (AMD Ryzen 9 7950X, GeForce RTX 3090, 64 GB RAM) and took approximately 20 hours.

### 3.3.2 GEOMETRY IMAGE CREATION

To create Multi-Chart Geometry Images for 3D meshes, we use their existing UV maps. UV mapping is defined as the mapping $\rho : V \to [0, 1]^2$, where $V$ is the set of vertices in a 3D mesh, and $\rho$ maps each

vertex to UV coordinates. Note that this mapping is not injective (multiple vertices can be mapped to the same UV coordinates), nor is it a simple function of the vertex positions in $\mathbb{R}^3$, as modern mesh formats allow multiple vertices with distinct UV coordinates at the same 3D location. These issues mean that $\rho$ is not invertible, which would be a simple way to create a geometry image function. However, we argue that $\rho$ is *locally invertible* and propose constructing a Multi-Chart Geometry Image based on the individually invertible areas of the available UV mapping. As mentioned before, the charts in a UV atlas tend to be semantically meaningful, so we aim to preserve them.

We begin by identifying the connected components of the mesh, which provides an initial separation into charts. Within each component, we identify two situations where $\rho$ is not invertible: duplicated vertices with distinct UV coordinates, and a "crease", i.e. a line where the UV coordinates change direction (similar to a "mirror" boundary condition). The former is straightforward to detect, as we can find duplicated vertex positions with different UV coordinates. The second case is identified by creating a heatmap of the UV-space access pattern and detecting local minima, which indicate an indexing pattern that "doubles back" on itself. We then further split the individual charts along any detected creases. In line with the desirable mapping properties discussed in (Sinha et al., 2016), we adjust the geometry image mapping to approximate an equal-area projection by rescaling each 2D chart with respect to the area of the corresponding surface.

In cases where only a partial UV mapping is available, we use XAtlas (Young, 2022) to UV-unwrap the missing regions. However, since the XAtlas parametrization is of lower quality and lacks semantic properties, we exclude meshes where less than $80\%$ of the surface area has been unwrapped manually. This simple heuristic allows us to construct multi-chart geometry images for nearly all training examples. However, this method does not account for all possible degenerate cases of $\rho$. Therefore, we verify that the constructed $\rho$ is injective and skip any training meshes where this assumption is violated.

In Objaverse (Deitke et al., 2022), the objects are not aligned in a canonical way. Although most shapes are oriented with the Y-axis pointing upward, there is no fixed front view, leading to rotational ambiguity between the X- and Z-axes. To resolve this ambiguity and confine it to a single coordinate, we leverage cylindrical coordinates $(r, \theta, \phi)$ so that the ambiguity is restricted to the azimuth angle $\theta$.

### 3.3.3 MESH EXTRACTION

As mentioned in section 3.1, geometry images implicitly encode connectivity information of the mesh by treating neighboring pixels as connected in 3D space, i.e. a quad mesh. However, to convert them into a more widely supported triangular mesh, we need to specify exactly how to triangulate the quads. For this, we closely follow the algorithm from Sander et al. (2003). For any $2 \times 2$ block of pixels in the GIM, we create up to two triangles depending on how many of the pixels describe part of the object. If necessary, the quad is split along its shorter diagonal. As

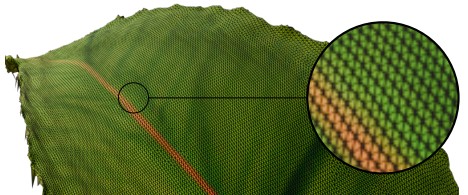

Figure 4: The resulting triangulation of our generated objects is nearly uniform across the surface, due to the area-preserving nature of the geometry images in our training dataset.

shown in fig. 4, and in line with our goal of area-preserving mapping (Sinha et al., 2016), the resulting triangulation is nearly uniform over the surface, which may or may not be desirable for specific applications. With our model's working resolution of $768 \times 768$, our GIMs can encode meshes with up to $589,824$ vertices. We consider the generation of arbitrary topologies or mesh generation with polygon constraints a promising area of future work for GIMDiffusion.

### 3.4 TRAINING

For the frozen base model, we used a zero-terminal-SNR (Lin et al., 2024) version fine-tuned from StableDiffusion v2.1 (Rombach et al., 2021) as the base Text-to-Image model, which remains frozen and generates albedo textures. The geometry model is an architectural clone with the cross-attention layers omitted and is trained from scratch, along with the cross-network communication layers. Initially, we trained the model at $256 \times 256$ resolution for $250,000$ steps with a batch size of $384$, and then at the final output resolution of $768 \times 768$ for a total of $100,000$ steps with a batch size of $64$. All stages of training were conducted with a learning rate of $3e^{-5}$ on 8 A100 GPUs.

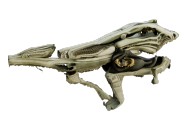 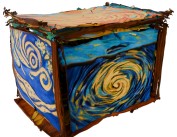 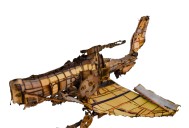 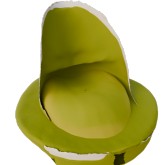 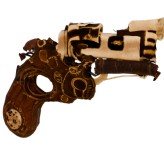

*An organic alien gun is style of H.R. Giger* *Wooden chest with Van Gogh's Starry Night* *A steampunk airplane* *An avocado-shaped chair* *A steampunk gun*

Figure 5: GIMDiffusion generalizes well beyond the training data, successfully following more outlandish prompts.

# 4 RESULTS

Figure 1 showcases the results of our method on a set of text prompts, generating objects in a manner that aligns with common queries, such as those in a gaming workflow. The objects are **well-defined** and can be **relit from any direction**, as the albedo textures are free from lighting-related artifacts. The generated UV layouts closely resemble those created by artists, offering editable assets.

In fig. 6 we highlight the method's ability to generate meaningful variations when the seed or prompt is perturbed. GIMDiffusion produces significantly different UV layouts with even slight variations in the seed or prompt, enabling diverse object creation. Leveraging the rich natural image prior, our model generalizes well beyond the initial 3D dataset (see appendix E for the layout novelty study). Further examples illustrating this generalization are shown in fig. 5.

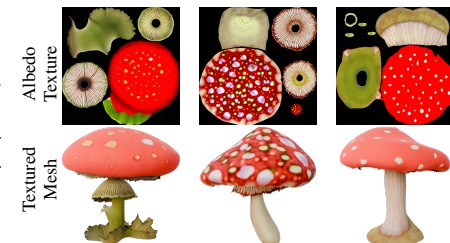

**Prompt:** *A fly agaric mushroom*

Figure 6: Sample diversity of GIMDiffusion for different random seeds.

## 4.1 SEPARABILITY AND INTERNAL STRUCTURE

A key advantage of our method is its ability to generate objects divided into distinct semantic (or nearly semantic) parts, as shown in fig. 7, making the generated objects more suitable for editing and animation. This capability arises from the multi-chart representation design and the semantic information embedded in the training data through handcrafted UV maps, which loosely correspond to the semantic components of objects. This approach also allows users to easily correct imperfections, such as misaligned parts or extraneous geometry, and even combine different parts from multiple generations. Additionally, our method generates internal structures, such as the filament inside a light bulb or the interior of a fish tank, because geometry images represent the entire object holistically, not just its visible surfaces.

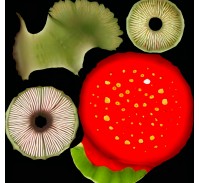 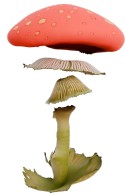

*Albedo texture* *Exploded view*

Figure 7: An exploded view of the generated "Fly agaric mushroom" shows semantically meaningful chart separation.

## 4.2 IPADAPTER COMPATIBILITY

Major efforts have been made to achieve style control in diffusion models (Ye et al., 2023), and our method is compatible with these techniques. Since we leverage a frozen base model to generate the albedo textures, we can apply a pre-trained IPAdapter and produce stylized output meshes, as shown in fig. 8. Despite the significant mismatch between the application domains (natural images versus albedo atlases), the style guidance remains successful. However, we find that this approach starts to break when the text prompt deviates too much structurally from

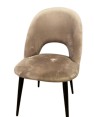 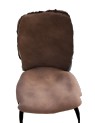 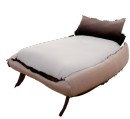

*IPAdapter input* *<No prompt>* *A kingsize bed*

Figure 8: We can guide the reverse process by applying a pre-trained IPAdapter to the frozen base model (Ye et al., 2023).

the content of the conditioning image. We attribute this to the fact that IPAdapter aims to leverage every aspect of the conditioning image; but e.g. IPAdapterInstruct (Rowles et al., 2024) offers a selective extraction of just appearance style without entangling structural information.

## 4.3 EVALUATION

*A cactus with pink flowers*

*A pair of polka-dotted sneakers*

*An antique gold pocket watch*

*An old, frayed straw hat*

*A worn-out leather briefcase*

| MVDream | GaussianDreamer | LGM | CRM | SF3D | Ours |

Figure 9: The qualitative evaluation on prompts from T$^3$Bench (He et al., 2023). Note that our model produces well-defined shapes and detailed appearances due to high-resolution of underlying geometry images. The only noticeable defects are visible cracks and minor misplacements of small parts, which we aim to address in future work (see section 4.5).

We selected several recent works with available source code for comparison, including both optimization-based and feed-forward methods. Specifically, we chose GaussianDreamer (Yi et al., 2023), DreamGaussian (Tang et al., 2023), MVDream (Shi et al., 2023b), LGM (Tang et al., 2024), CRM (Wang et al., 2024) and SF3D (Boss et al., 2024). We adapted state-of-the-art Image-to-3D models LGM, CRM and SF3D for Text-to-3D by integrating the same Text-to-Image model used in our approach (see more details on evaluation in appendix C).

We treated the 3D mesh as the final output[1] and included post-processing time in the total generation time, using default settings for all methods. For Image-to-3D models, we also accounted for

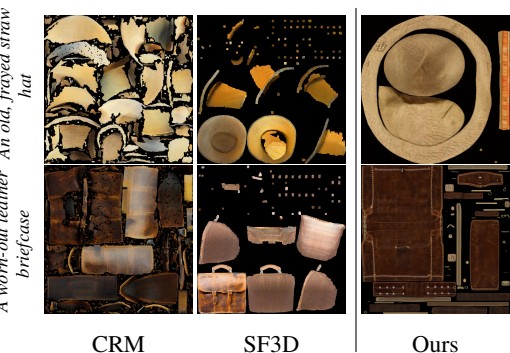

*An old, frayed straw hat*

*A worn-out leather briefcase*

| CRM | SF3D | Ours |

Figure 10: Our method generates layouts of higher quality than parametrization algorithms used in other methods.

---

[1]GaussianDreamer outputs Gaussian splats, which we converted using the method provided by LGM for evaluation

Table 1: Evaluations

| Method | Time | UV-unwrapping | Relightable | Separable | $T^3$ ↑ | AR > 4 (%) ↓ | RR > 4 (%) ↓ | MA < 10 (%) ↓ |
|---|---|---|---|---|---|---|---|---|
| GaussianDreamer | 15 minutes | No | No | No | 28.7 | - | - | - |
| MVDream | 30 minutes | XAtlas | No | No | 47.8 | - | - | - |
| DreamGaussian | 7 minutes | XAtlas | No | No | 19.8 | - | - | - |
| LGM | 1.8 minutes | XAtlas | No | No | 29.9 | 0.068 | 0.081 | 0.060 |
| CRM | 13 sec | XAtlas | No | No | 39.6 | **0.012** | **0.017** | **0.013** |
| SF3D | **3.5 sec** | Cube projection | **Yes** | No | **48.2** | 0.112 | 0.117 | 0.122 |
| Ours | 10 sec | **Generative** | **Yes** | **Yes** | 35.2 | 0.038 | 0.042 | 0.035 |

the 3-second generation time required by our base model to create a single image. All evaluations were performed on a single A100 GPU.

We used T³Bench (He et al., 2023) for automatic evaluation, measuring both generation quality and prompt alignment of the resulting meshes. T³Bench includes three prompt sets: Single Object, Single Object with Surroundings, and Multiple Objects. Since our model is specifically designed for single-object generation, we focused our evaluation on that track. We used the code provided in the T³Bench repository[2] for comparisons. Overall, we evaluated 100 prompts (see appendix D). Visual comparisons are provided in fig. 9.

T³Bench evaluates only multi-view renderings, it does not assess the underlying mesh quality. To address this, we followed Shen et al. (2023) and calculated metrics such as the percentage of triangle aspect ratios > 4 (**AR>4(%)**), radius ratios > 4 (**RR>4(%)**), and minimum angles < 10 (**MA<10(%)**), using PyVista (Sullivan & Kaszynski, 2019) for all feed-forward methods. While these intrinsic metrics cannot guarantee overall quality, they provide reasonable proxy for some applications.

As shown in table 1, our method achieves competitive results compared to state-of-the-art models, while providing superior editability via separable parts and producing nearly artistic UV maps. Unlike other approaches that rely on vertex colors or pre-defined fixed unwrapping, our method natively generates UV-space textures. This results in more logical chart separation, better area utilization, and sharper details compared to other methods, as demonstrated in fig. 10.

## 4.4 ABLATION STUDY

In this section, we validate our method by ablating key design choices and evaluating their impact. Specifically, we ablated three aspects of our approach: the absence of cross-attention layers in the geometry image branch, the use of the Collaborative Control mechanism, and the cylindrical coordinate transform. To ensure consistency, all experiments used the same batch size, number of steps, and hardware.

First, we reintroduced **cross-attention layers** in the geometry image branch, making it a full architectural clone of the frozen branch. As shown in the second row of fig. 11, while the model still follows the prompts, its generalization ability significantly deteriorates, especially with out-of-distribution prompts like "a steampunk gun", which is consistent with the observations in Vainer et al. (2024).

Next, we removed the **Collaborative Control** mechanism and fine-tuned the base model to generate both albedo textures and geometry images, concatenating their latents. This resulted in a significant drop in both shape and texture quality, as seen in the third row of fig. 11.

Finally, we ablated **cylindrical coordinates** by fine-tuning the model as before but omitting the coordinate transformation. Without this step, the model struggled to generate even basic shapes.

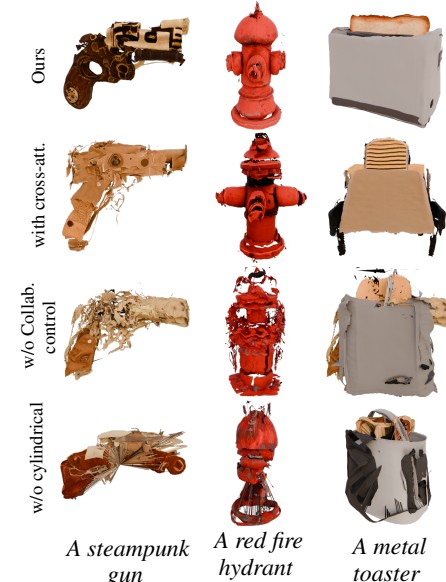

*A steampunk gun*  *A red fire hydrant*  *A metal toaster*

Figure 11: The first row shows our full method's performance, with subsequent rows removing key design elements.

---
[2]https://github.com/THU-LYJ-Lab/T3Bench

## 4.5 LIMITATIONS

Our current method has certain limitations. The most common issue is the appearance of visible cracks in the generated meshes. These artifacts result from two key factors: Multi-Chart Geometry Images lack inter-chart connectivity information, and irregular chart boundaries are prone to undersampling (see fig. 12), as analyzed by Sander et al. (2003). The original paper proposed a zippering algorithm to close the cracks between adjacent charts, but this approach assumes the watertightness of the underlying mesh, which is not generally true for real-world meshes. In future work, we aim to generalize and adapt the zippering algorithm. Sampling-related modifications, such as those proposed by Gauthier & Poulin (2009); Yan et al. (2024), could further improve the results.

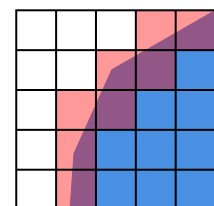

Figure 12: The blue area represents a chart, undersampling at insufficient resolutions can affect chart boundaries. Affected pixels are highlighted in red.

Additionally, we believe this issue is exacerbated by the VAE's latent compression. Areas smaller than $8 \times 8$ pixels fall below the VAE's latent resolution, leading to visual artifacts such as noisy edges and misaligned charts. We hypothesize that integrating wavelets into the VAE, as suggested by Sadat et al. (2024), could mitigate these issues.

Another limitation stems from the discrepancy between the natural image prior and the domain of our model. Although individual charts in a geometry image can be arbitrarily rotated and still represent the same 3D object, the frozen base model is not rotationally equivariant (Weiler et al., 2021). Human faces serve as a good example — while typically upright in natural images, they can appear in random orientations on texture maps, leading to varying quality for such prompts (see examples in appendix B). In future work, we plan to investigate ways to establish canonical orientations for these charts, ensuring better alignment with the diffusion prior.

Additionally, we sometimes observe that the model duplicates parts of the object, which can result in visual artifacts in the generated meshes due to z-fighting. However, the segmented nature of the generated objects makes these artifacts easy to resolve manually.

## 5 DISCUSSION AND FUTURE WORK

In this work, we present **Geometry Image Diffusion** (GIMDiffusion), a novel Text-to-3D generation paradigm that utilizes geometry images as its core 3D representation in combination with powerful natural image priors in the form of pre-trained diffusion models. Our results show that GIMDiffusion can generate relightable 3D assets with high-quality UV maps as efficiently as existing Text-to-Image methods generate normal images, while avoiding the need for complex, custom 3D-aware architectures. We believe that our research lays the groundwork for a new direction in Text-to-3D generation.

Future improvements include eliminating visible cracks, achieving better alignment with 2D priors, and enhancing the precision of generations. Incorporating topology prediction and conditioning on specific polygon budgets could provide greater control over the generated 3D objects, making them more practical for gaming and other graphics pipelines. Furthermore, the inherent capability of GIMDiffusion to model both geometry and parameterization makes it particularly well-suited to addressing the longstanding challenge of UV mapping creation for existing geometries. Equally promising is the potential of GIMDiffusion in related fields such as interactive editing, animation and Text-to-Video generation.

### ACKNOWLEDGMENTS

We would like to thank Shimon Vainer from Unity Technologies, Dr. Lev Melnikovsky from the Weizmann Institute and Alexander Demidko for their valuable feedback and insightful discussions. Konstantin Kutsy has been invaluable in helping us operate our training infrastructure.

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

## A    VAE DESIGN AND TRAINING DETAILS

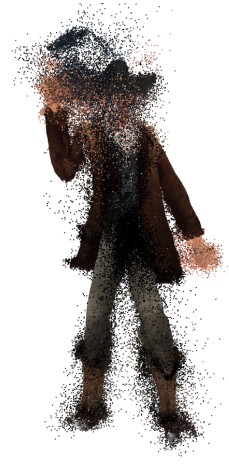

SD VAE reconstruction

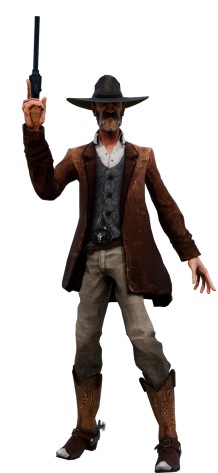

GIM VAE reconstruction

Figure 13: Comparison of geometry reconstruction between the SD VAE and our GIM VAE. The SD VAE reconstruction could not be meaningfully triangulated, a point cloud is visualized instead of a mesh

The Stable Diffusion Variational Autoencoder (SD VAE) is trained on natural images and achieves high reconstruction quality in that domain. However, it does not transfer well to the domain of geometry images.

The first issue is that the reconstructed images contain a considerable amount of noise (see fig. 13). While these perturbations may be imperceptible in the image domain, they cause significant problems in the 3D domain, such as the loss of fine geometric details. We hypothesize that this behavior arises from the LPIPS (Zhang et al., 2018) and GAN losses used during training, as GANs are known for introducing high-frequency noise into images.

The second issue is the color range bias of the SD VAE, which favors the most common colors in natural images. When applied to points in 3D space, this bias can cause undesirable warping and other artifacts.

Lastly, the original VAE struggles to accurately reconstruct discontinuities at chart boundaries, which is critical for triangulation (see section 3.3.3). Convolutional layers tend to interpolate across boundaries, leading to false geometry due to the creation of smooth transitions instead of sharp ones.

Table 2: VAE evaluations

| Method | Latent dim | PSNR ↑ |
|--------|-----------|--------|
| SD VAE | $96 \times 96 \times 4$ | 31.87 |
| GIM VAE | $96 \times 96 \times 8$ | 49.02 |

To address these issues, we made the following design choices:

- **Disentangling chart boundaries from geometry:** We introduced additional input and output channels for a multi-chart mask. This mask can be thresholded to model sharp boundaries, avoiding interpolation artifacts.

- **Removing GAN and perceptual losses:** These were removed to prevent noise from being introduced during training.

- **Increasing latent dimension:** We expanded the latent dimension to 8 to better capture geometric details.

We trained our VAE following the procedure and codebase[3] of Stable Diffusion (Rombach et al., 2021). The model was trained on 8 A100 GPUs with a batch size of 128 for 100k steps. The PSNR results are reported in table 2.

## B  FAILURE CASES

Charts in the Geometry Image layout can be randomly rotated without affecting the resulting 3D shape, as this rotation alters only the parameterization. However, GIMDiffusion relies on Stable Diffusion 2.1 as a prior, which is not rotationally equivariant and exhibits strong orientational biases for certain features. For example, it struggles to reliably generate an upside-down face. Consequently, the quality of generated outputs may degrade for certain prompts. With the prompt *"human head"*, for instance, the quality can vary depending on the seed, particularly when the chart containing the face is oriented unfavorably.

Fortunately, GIMDiffusion's fast inference speed enables efficient seed exploration to achieve the desired quality. In future work, we aim to introduce canonical orientations for charts to mitigate this issue.

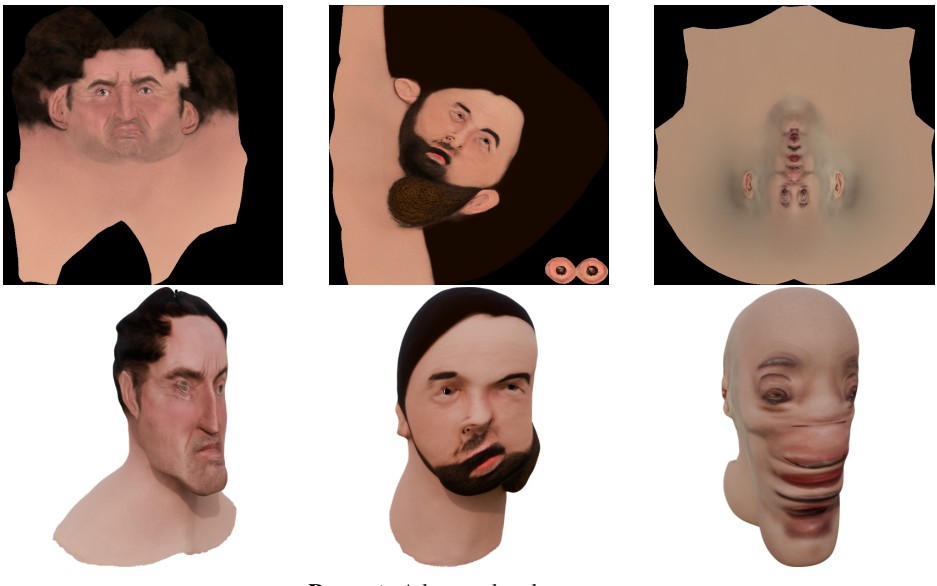

**Prompt:** *A human head*

Figure 14: Generation results for *human head* for three different seeds.

## C  EVALUATION DETAILS

$T^3$Bench (He et al., 2023) evaluates both quality and alignment in text-to-3D generation. Quality is assessed through a standardized multi-view rendering pipeline provided by the authors. This process generates 2D images of the 3D object from various angles and uses scoring models (Radford et al., 2021; Xu et al., 2024) to evaluate overall visual consistency and fidelity. To spot issues such as view inconsistency (e.g., the Janus problem), a regional convolution technique is applied. Alignment measures the semantic consistency between the generated 3D object and the input text prompts using a multi-view captioning process coupled with GPT-4 (Achiam et al., 2023) evaluation. The final score is computed as the average of the quality and alignment metrics.

To compare our method with state-of-the-art image-to-3D approaches, we generated a set of images based on the $T^3$Bench evaluation prompts (see appendix D) using the same base model as our method (Stable Diffusion 2.1). For compatibility with image-to-3D methods, we appended the phrase *"single object, uniform background"* to each prompt. For each prompt, three images were generated, and the

---

[3]https://github.com/Stability-AI/generative-models

one best aligning with the text and depicting the object from natural angles was manually selected. Figure 15 showcases several examples.

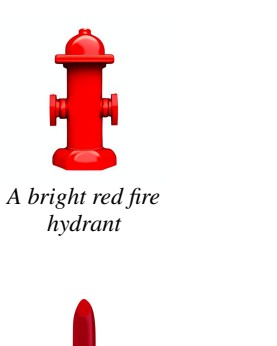
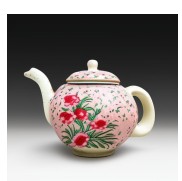
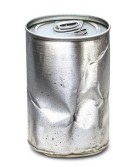

*A bright red fire hydrant*

*A ceramic teapot with floral patterns*

*A worn-out leather briefcase*

*A crumpled silver aluminum soda can*

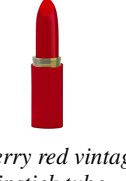
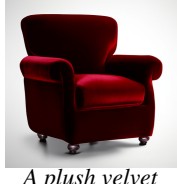
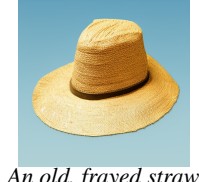
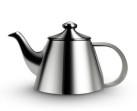

*A cherry red vintage lipstick tube*

*A plush velvet armchair*

*An old, frayed straw hat*

*A sleek stainless steel teapot*

Figure 15: Examples of images used to condition Image-to-3D models during evaluation.

## D  EVALUATION PROMPTS

1. A cactus with pink flowers
2. A rainbow-colored umbrella
3. An antique wooden rocking horse
4. A golden retriever with a blue bowtie
5. An ivory candlestick holder
6. A pair of polka-dotted sneakers
7. A steaming mug of hot chocolate with whipped cream
8. A bright red fire hydrant
9. A gleaming silver saxophone
10. A leather-bound book with gold details
11. A vibrant sunflower with green leaves
12. A castle-shaped sandcastle
13. A neon green skateboard with black wheels
14. A pirate flag with skull and crossbones
15. A plush teddy bear with a satin bow
16. A ripe watermelon sliced in half
17. A sparkling diamond ring in a velvet box
18. A vintage porcelain doll with a frilly dress
19. A chameleon perched on a tree branch
20. A tarnished brass pocket watch
21. A ceramic teapot with floral patterns
22. An antique ruby-studded brooch
23. A simple burgundy colored feather quill
24. A vintage iron-cast typewriter
25. A shiny emerald green beetle
26. A crystal glass paperweight with abstract design
27. A velvet cushion stitched with golden threads
28. A small porcelain white rabbit figurine
29. A left-handed electric guitar painted black
30. A bright blue plastic swimming goggles
31. A partly broken shell of a tortoise
32. A long woolen scarf, striped red and black
33. A tattered old explorer's map
34. A well-used black iron frying pan
35. A crumpled silver aluminum soda can
36. A thick, green-spined book with yellowed pages
37. A shimmering emerald pendant necklace
38. A well-worn straw sun hat
39. A tarnished silver letter opener
40. An antique glass perfume bottle
41. A polished mahogany grand piano
42. A dented brass trumpet
43. A pristine white wedding gown
44. A chipped porcelain teacup
45. A rustic wrought-iron candle holder
46. A vibrant, handmade patchwork quilt
47. A plush velvet armchair
48. A sleek, black top hat
49. A paint-splattered easel
50. A bent steel crowbar
51. A crisp paper airplane
52. A worn-out rubber tire swing
53. An intricately-carved wooden chess set
54. A bright red kite with a frayed tail
55. A smooth, round opal stone
56. A rusty, vintage metal key
57. A delicate, handmade lace doily
58. A sturdy mahogany walking cane
59. A sparkling crystal chandelier
60. A worn-out red flannel shirt
61. A cracked porcelain doll's face

62. A dusty classic typewriter
63. A glossy grand black piano
64. A faux-fur leopard print hat
65. A futuristic, sleek electric car model
66. A cherry red vintage lipstick tube
67. A cobweb-covered old wooden chest
68. A gold glittery carnival mask
69. A tattered world map with stained edges
70. A shiny red apple
71. A worn-out leather briefcase
72. An antique gold pocket watch
73. A sleek, slim smartphone
74. A wet, vibrant beach ball
75. A rusty, abandoned bicycle
76. A fluffy, orange cat
77. Crisp, folded origami paper
78. A shiny, new electric guitar
79. A weather-beaten wooden bat
80. A delicate crystal champagne flute
81. An old, frayed straw hat
82. A scuffed up soccer ball
83. A pair of worn-in blue jeans
84. A well-loved stuffed teddy bear
85. A chipped, white coffee mug
86. A bright, yellow rubber duck
87. A sleek stainless steel teapot
88. A water-streaked glass window pane
89. An intricate ceramic vase with peonies painted on it
90. A fuzzy pink flamingo lawn ornament
91. A blooming potted orchid with purple flowers
92. An old bronze ship's wheel
93. A sparkling diamond tiara
94. A vintage plaid woolen blanket
95. A pair of shiny black patent leather shoes
96. An elegant feather-quill ink pen
97. A fragrant pine Christmas wreath
98. A silver mirror with ornate detailing
99. A green enameled watering can
100. A classic leatherette radio with dials

# E  LAYOUT NOVELTY

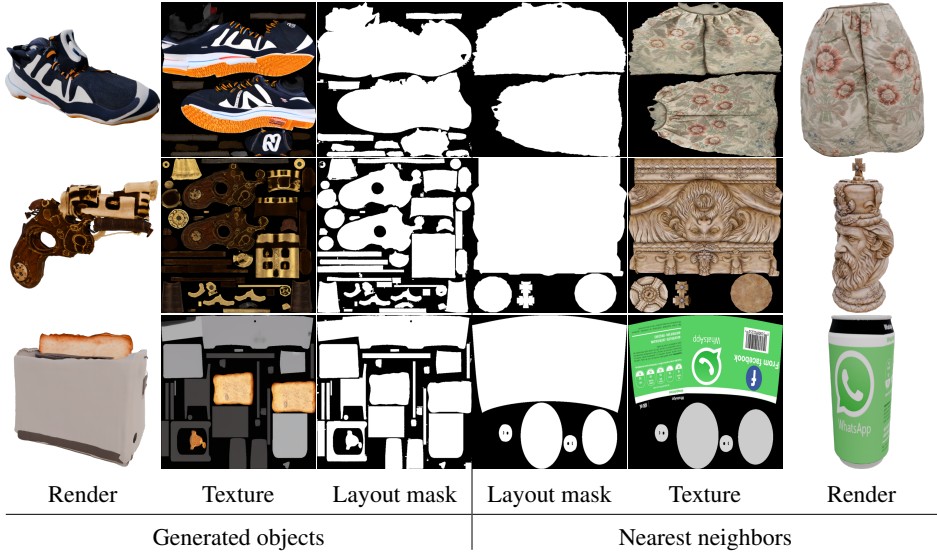

| Render | Texture | Layout mask | Layout mask | Texture | Render |
|---|---|---|---|---|---|
| | Generated objects | | | Nearest neighbors | |

Figure 16: Each row displays a generated object alongside its nearest neighbor by layout mask from the dataset. The nearest UV layouts differ significantly from the corresponding generated ones, often representing entirely different object types.

To demonstrate our model's ability to generate novel UV layouts not present in the training data, we visualize dataset entries that are closest to the generated samples in terms of layout similarity. For this purpose, we constructed an index of binary multi-chart masks extracted from all geometry images in our dataset using the Faiss library (Douze et al., 2024). We then queried this index to find the nearest neighbors for the generated samples.

During the training phase of our model, the input data were augmented with random rotations by angles that are multiples of 90 degrees. Accordingly, for each query image, we computed the nearest neighbors across four rotations and selected the neighbor with the lowest Euclidean distance.

The visualizations in fig. 16 reveal that there are non-trivial differences between the generated layouts and those present in the training data, highlighting the model's capability for generating novel layouts.

