# OpenReview forum: "Geometry Image Diffusion: Fast and Data-Efficient Text-to-3D with Image-Based Surface Representation"
_ICLR.cc/2025/Conference — ICLR 2025 Poster_

### Official Review · Reviewer_tGfn · 2024-10-16

**Soundness:** 3
**Presentation:** 3
**Contribution:** 3
**Rating:** 6
**Confidence:** 4

**Summary:**

This paper presents a method to generate geometry images with diffusion models, which represents meshes with texture maps with images. Specifically, the authors finetuned stablediffusion to generate the albeda images and use collaborative control to generate the geometry images, which can then be unfolded to 3D meshes. The authors evaluated their method on T^3 bench and shows that it can generally well beyond training data and also have better geometry.

**Strengths:**

- The idea of using pretrained diffusion models for native 3D generation is interesting and novel
- The proposed method is straightforward and easy to understand. The proposed components are also well validated in the ablation study.
- The generated 3D meshes have semantic information and can be decomposed into several parts, make it potential for downstream editing tasks. It also directly generates UV maps and is directly relightable.

**Weaknesses:**

- Although the generated meshes are quite sharp and have details, there are obvious discontinuous artifacts, which can be observed in the video. The reason might be that during reconstructing the ground truth geometry images, these parts are already disconnected. Also, in Fig. 7, the boundaries of the mesh also have noticeable artifacts. What is the reason behind these artifacts? Are these artifacts solvable by improving the current pipeline? Is it possible to apply some post-processing techniques to deal with that?
- The details of the evaluation is not that clear: how many prompts are used during evaluation and what are they? The authors may provide the number of prompts and some examples of that.
- I would like to see more comparison with two-stage mesh methods, e.g. first multi-view and then reconstruct mesh, especially CRM, which works faster and have better quality than LGM to my knowledge. It can also produces UV maps. Besides, the authors should also show more examples of comparison in the paper. It would be great if the authors could provide the same comparisons similar to what have been shown in Table 1 for CR.

**Questions:**

I noticed that the albedo branch used the pretrained VAE, buth GIM is trained from scratch. Have the authors considered also using the pretrained model for GIM? What is the reason of not using the pretrained model? Is there any comparison for that?

---

> ### Author Response · Authors · 2024-11-24
>
> We sincerely thank the reviewer for their thoughtful comments, highlighting the novelty and strengths of our work, as well as for the constructive feedback. Below, we address the points raised in detail:
>
> ### **Analysis of Pretrained VAE**
> The Stable Diffusion VAE, while effective for natural images, encounters several challenges when applied to geometry images:
> 1. **Noise in Reconstructions**: The VAE introduces high-frequency noise, likely due to LPIPS and GAN losses, which severely degrade fine 3D geometric details (see Figure 13).
> 2. **Color Bias**: Its bias toward natural image color palettes results in undesirable warping when mapped to 3D space.
> 3. **Boundary Handling**: The SD VAE struggles to reconstruct sharp discontinuities at chart boundaries, which are essential for accurate triangulation.
>
> We have updated our paper to include an analysis of the VAE's behavior on GIMs, the rationale behind our design choices, and a report of PSNR for both VAEs (see Appendix A).
>
> ---
>
> ### **Cracks and Boundary Artifacts**
> The cracks and boundary artifacts arise from two key factors:
> 1. **Lack of Inter-Chart Connectivity**: Multi-chart geometry images do not inherently capture connectivity information between charts, resulting in disconnected parts.
> 2. **Irregular Chart Boundaries**: These boundaries are prone to undersampling, leading to visible cracks, as analyzed in the original MGIM paper [1].
>
> In the original MGIM paper, a zippering algorithm was proposed, but it assumes that the mesh is watertight, which is not guaranteed in real-world datasets. We plan to generalize and adapt this algorithm in future work. Additionally, sampling modifications and further VAE improvements could help mitigate the problem.
>
> Further analysis and visualizations are provided in Section 4.5.
>
> ---
>
> ### **Evaluation**
> Thank you for raising concerns about the evaluation details. We have updated our paper to include:
> - A complete list of prompts used during evaluation (Appendix D).
> - Details on adapting Image-to-3D models for Text-to-3D generation (Appendix C).
>
> We provided both quantitative and qualitative evaluations of the CRM method:
> 1. **Quantitative**: On the T³ Bench, CRM achieved a score of 39.6, compared to 35.2 for our model (see Table 1 for all results).
> 2. **Qualitative**: Figure 9 includes visual comparisons. Figure 10 compares CRM's UV layouts (XAtlas) with our model's generative UV layouts.
>
> We hope these updates provide a clearer and more comprehensive understanding of our evaluation process and comparisons.
>
> ---
>
> We kindly ask the reviewer to reconsider their score based on the updated submission.
>
> ---
>
> ### **References**
> [1] Sander P, Wood Z, Gortler S, Snyder J, Hoppe H. Multi-Chart Geometry Images. Eurographics Symposium on Geometry Processing, 2003. [Available here](https://api.semanticscholar.org/CorpusID:4893489).

---

> > ### Comment · Reviewer_tGfn · 2024-11-25
> >
> > I would like to thank the authors for their rebuttal, which addresses most of my concerns. I would like to raise my score.

---

### Official Review · Reviewer_zGeZ · 2024-10-31

**Soundness:** 3
**Presentation:** 3
**Contribution:** 2
**Rating:** 6
**Confidence:** 4

**Summary:**

1. GIMDiffusion uses geometry images to represent the surface of 3D shapes in a 2D format. This method, unlike traditional 3D meshes, can utilize the extensive 2D image generation capabilities of diffusion models like Stable Diffusion. This approach results in faster generation times, lower data requirements, and the elimination of the need for complex 3D-specific model architectures, which are often cumbersome and data-hungry.

2. The model introduces a Collaborative Control mechanism to integrate 2D Text-to-Image models with geometry images. Through this mechanism, the geometry and texture generation processes are interconnected, allowing each to influence the other. This setup ensures that the textures generated align well with the 3D shape, facilitating realistic and coherent representations even when there is limited 3D data available.

3. Efficiency and Performance: GIMDiffusion achieves faster-than-average generation times, producing complex 3D assets in approximately 10 seconds. Moreover, the model avoids intermediate 3D processing stages like iso-surface extraction, further streamlining the pipeline.

**Strengths:**

1. Innovative 2D-Based Approach for 3D Generation: By using geometry images, GIMDiffusion avoids the need for specialized 3D architectures, simplifying the overall system and reducing the computational resources required.
2. The paper introduces a Collaborative Control mechanism, which allows two parallel diffusion models to communicate effectively. This cross-model communication lets the geometry generation model inform and guide the 2D texture model, a unique architectural contribution that enhances model efficiency and adaptability.
3. Rapid and Resource-Efficient Processing: The generation time is highly competitive, making this model a viable choice for real-time applications and scalable deployment.

**Weaknesses:**

1. In figure 9, the paper observes artifacts, such as visible seams between geometry images, which could compromise visual quality, especially in high-precision applications requiring seamless textures and continuous surfaces.

2. While the paper highlights the efficiency and practicality of geometry images, it doesn’t fully address potential limitations of this 2D representation for complex or high-genus shapes. Multi-chart geometry images mitigate some challenges, but further analysis is needed on cases where this approach may fail, such as in scenarios with high geometric distortion or disconnected regions.

3. Limited Comparison with Recent Methods: The paper's evaluation lacks comprehensive comparisons with the latest advancements, instead focusing on older benchmarks like DreamFusion, ProlificDreamer, and Shap-E. Expanding comparisons to include recent approaches, such as MVDream, CRM, DreamGaussian, and Text-to-3D using Gaussian Splatting, would clarify where GIMDiffusion specifically excels or encounters limitations.

**Questions:**

1. Could you include comparisons with more recent Text-to-3D approaches, such as MVDream, CRM, DreamGaussian, and other fast, high-quality methods based on Gaussian Splatting? Including comparisons to more recent approaches would offer readers a clearer understanding of where GIMDiffusion stands relative to current advancements.

2. How does GIMDiffusion perform with diverse object types, especially those with intricate internal structures or shapes? Could you provide more detail on potential methods to mitigate visible seams or artifacts between geometry images? It would be helpful if you could discuss or experiment with approaches for reducing these seams, such as stitching algorithms or blending techniques that could improve texture continuity, especially for high-fidelity applications.

3. The visual quality of this method is not obviously better than some other recent methods, which raises concerns about its competitive edge. While GIMDiffusion offers efficiency and compatibility with 2D architectures, it would benefit from additional improvements or evaluations to enhance the clarity, realism, and continuity of generated textures and geometry.

---

> ### Author Response · Authors · 2024-11-24
>
> We thank the reviewer for their thoughtful comments, for highlighting the strengths of our work, and for providing constructive feedback.
>
> We appreciate the reviewer’s recognition of our adaptation of the Collaborative Control mechanism. However, we would like to kindly clarify that Collaborative Control was initially introduced in [1] for PBR material generation. In our work, this mechanism has been re-used and adapted for geometry images to enable effective integration of geometry and texture generation.
>
> ---
>
> ### **Evaluation**
>
> We have added new evaluations of **MVDream**, **DreamGaussian**, and **CRM**, complementing the existing comparisons with **GaussianDreamer**, **LGM**, and **SF3D** (see Section 4.3). The visual comparisons have also been updated and expanded, with additional results presented in Figure 9. Furthermore, Figure 10 provides a direct visual comparison of UV-space textures produced by our method and competing approaches.
>
> The results demonstrate that **GIMDiffusion** is competitive with state-of-the-art methods on the **T³ Bench** [4].
>
> ---
>
> ### **Artifacts and Limitations**
>
> We acknowledge the reviewer’s concerns about visible seams and potential limitations in the representation. In response, we have expanded the analysis of our method’s limitations, detailing the sources of boundary artifacts and proposing potential solutions (see Section 4.5 and Appendix B).
>
> #### **Key Findings**:
> 1. **Lack of Inter-Chart Connectivity**: Multi-chart geometry images do not inherently encode connectivity information between charts, leading to discontinuities.
> 2. **Irregular Chart Boundaries**: These boundaries are prone to undersampling, resulting in visible cracks, as analyzed by [2].
>
> In the original MGIM paper, a zippering algorithm was proposed, but it assumes that the mesh is watertight, which is not guaranteed in real-world datasets. We plan to generalize and adapt this algorithm in future work. Additionally, sampling modifications and further VAE improvements could help mitigate the problem.
> Further analysis and visualizations are provided in Section 4.5.
>
> ---
>
> We sincerely thank the reviewer for their feedback. We hope the additional evaluations and expanded limitation analysis address the concerns raised. We kindly ask the reviewer to reconsider their score in light of these updates.
>
> ---
>
> ### **References**
>
>
> [1] Vainer, Shimon, et al. "Collaborative control for geometry-conditioned pbr image generation." European Conference on Computer Vision. Springer, Cham, 2025.
> [2] Sander P, Wood Z, Gortler S, Snyder J, Hoppe H. Multi-Chart Geometry Images. *Eurographics Symposium on Geometry Processing*, 2003. [Available here](https://api.semanticscholar.org/CorpusID:4893489).
> [3] Gauthier M, Poulin P. Preserving Sharp Edges in Geometry Images. *Graphics Interface*, 2009.
> [4]  Bench: Benchmarking Current Progress in Text-to-3D Generation." arXiv preprint arXiv:2310.02977 (2023).

---

> ### Author Response · Authors · 2024-12-02
>
> As the discussion period comes to a close, we wanted to kindly follow up on our latest response, which includes new experiments and clarifications. We would greatly appreciate it if the reviewer could share any remaining concerns or insights before the discussion ends.
>
> If the response addresses the reviewer’s concerns, we would also kindly ask them to reconsider their score.
>
> We would like to thank the reviewer once again for their thoughtful feedback and time

---

> > ### Comment · Reviewer_zGeZ · 2024-12-02
> >
> > I would like to thank the authors for their effort to address some of my concerns. I would like to raise my score.

---

> > > ### Author Response · Authors · 2024-12-02
> > >
> > > We thank the reviewer for their kind words and their intention to raise the score. We noticed that the score has not yet been updated and is still 5, and we would like to kindly check if this was an oversight

---

### Official Review · Reviewer_nxjJ · 2024-11-02

**Soundness:** 3
**Presentation:** 3
**Contribution:** 2
**Rating:** 6
**Confidence:** 4

**Summary:**

The paper introduces Geometry Image Diffusion (GIMDiffusion), a novel Text-to-3D generation model that uses geometry images to represent 3D shapes in a 2D format. This approach leverages the priors of pre-trained image diffusion models, allowing efficient 3D object generation without complex 3D-aware architectures. By retaining pre-trained weights, GIMDiffusion is compatible with tools like IP-Adapter, enabling flexible style control and broad applicability. The use of multi-chart geometry images further enhances the model’s ability to represent complex shapes, offering a scalable and versatile solution for text-driven 3D generation.

**Strengths:**

1. Geometry images provide a low-complexity 3D representation, allowing GIMDiffusion to fully leverage pre-trained 2D diffusion models for 3D generation.
2. The model preserves the weights of the pre-trained image diffusion model, making it compatible with plugins like IP-Adapter for style transfer and customization.

**Weaknesses:**

1. My primary concern with multi-chart geometry images is the variable layout configurations that a single 3D shape can produce. That means a single 3D shape may correspond to many different geometry images, potentially complicating the training process and impacting consistency across different shapes. Is any technique used to standardize the conversion from mesh to geometry image.
2. Generated results in the paper often show lower quality around shape edges, possibly due to the geometry image representation. This limitation affects the clarity and sharpness of complex geometries.
3. The model freezes the pre-trained Stable Diffusion (SD) model for generating albedo textures. However, SD has known limitations in UV space, and freezing this branch could compromise RGB quality. Providing experiments to validate this decision would help substantiate the model’s robustness.

**Questions:**

1. How does the variability in multi-chart geometry image layouts impact GIMDiffusion’s training stability and scalability?
2. The quality of shape edges appears to be suboptimal in the generated results. Is this a known limitation of the geometry image representation, and are there potential solutions to address this?
3. The authors freeze the albedo generation branch based on the pre-trained SD model, but SD’s performance in UV space is often limited. Could the authors provide experimental evidence to show that this decision does not compromise the quality of generated RGB textures?

---

> ### Author Response · Authors · 2024-11-24
>
> We sincerely thank the reviewer for their detailed feedback and thoughtful questions. Please find our responses to the individual comments below:
>
> ### **Effects of Variability in Multi-Chart Geometry Image Layouts**
>
> We thank the reviewer for mentioning this interesting question. To standardize the dataset, individual charts were rescaled based on their area in 3D space, following [2], to approximate an equal-area projection.  Initially, we hypothesized that additional standardization, such as chart alignment or fixed packing rule, would be required. However, this simple procedure proved sufficient for stable training.
>
>
> The fully trained model demonstrates robust generalization, producing diverse layouts for different seeds (see Figure 6) and handling out-of-distribution prompts (Figure 5). The method scored between LGM [3] and CRM [4] in evaluations on a diverse set of prompts,  showing its ability to produce consistent shapes with sharp textures.
>
> Occasionally, the model generates extra charts (typically small parts), as noted in the last paragraph of Section 4.5. This may relate to the variability in layouts. Another limitation, discussed in Section 4.5 and Appendix B, is that the model struggles to generate patterns when a chart’s orientation does not align with the canonical orientation learned by the 2D prior.
>
> In broader terms, we do not view the many-to-many relationship between geometry and parameterizations as a limitation. Rather, we hypothesize that this allows the model to capture the joint distribution of shapes and UV maps, offering potential for various applications. For instance, with appropriate conditioning, the model could be utilized to generate UV maps for a given geometry or to generate geometry for a fixed UV layout. We plan to explore these possibilities in future work.
>
> ---
>
> ### **Effects of Freezing the SD Branch on Texture Quality**
>
> We experimented with different training strategies, as detailed in the ablation studies in Section 4.4. Fine-tuning the SD model led to overfitting, resulting in poor generalization and low-quality geometry and textures (see Figure 11). While fine-tuning may perform better with a significantly larger dataset, it was not effective in our setting.
>
> Our current approach, which freezes the SD branch, produces sharp and detailed textures (see Figures 9 and 10). This is largely due to the Collaborative Control mechanism, which efficiently utilizes and modifies the hidden states of the frozen diffusion model. This aligns with results by Vainer et al [5] in related context (PBR material generation).
> .
> A limitation of the SD branch is its lack of rotational equivariance, leading to orientation biases for certain features. If these features appear in unfavorable orientations in the texture, they may not render properly. Potential solutions to this issue are discussed in Section 4.5 and Appendix B.
>
> ---
>
> ### **Suboptimal Quality of Shape Edges**
>
> The cracks and boundary artifacts arise from two main factors:
> 1. **Lack of Inter-Chart Connectivity**: Multi-chart geometry images do not inherently encode connectivity information between adjacent charts, resulting in disconnected parts.
> 2. **Irregular Chart Boundaries**: These boundaries are prone to undersampling, leading to visible cracks, as analyzed in the original MGIM paper [1].
>
> The original MGIM paper proposed a zippering algorithm to address these issues. However, this approach assumes a watertight mesh, which is not guaranteed in real-world datasets. Future work will focus on generalizing and adapting this algorithm. Additionally, sampling modifications and further VAE improvements could help mitigate these problems.
>
> Further analysis and visualizations related to this issue are provided in Section 4.5.
>
> ---
>
> We greatly appreciate the reviewer’s feedback. The questions raised have helped us better articulate the strengths and limitations of our approach. We hope these updates address the reviewer’s concerns and kindly ask for reconsideration of the score.
>
> ---
>
> ### **References**
>
> [1] Sander P, Wood Z, Gortler S, Snyder J, Hoppe H. Multi-Chart Geometry Images. Eurographics Symposium on Geometry Processing, 2003.
>
> [2] Ayan Sinha, Jing Bai, and Karthik Ramani. Deep learning 3d shape surfaces using geometry images. In European Conference on Computer Vision, 2016
>
> [3] Tang, Jiaxiang, et al. "Lgm: Large multi-view gaussian model for high-resolution 3d content creation." European Conference on Computer Vision. Springer, Cham, 2025.
>
> [4] Wang, Zhengyi, et al. "Crm: Single image to 3d textured mesh with convolutional reconstruction model." European Conference on Computer Vision. Springer, Cham, 2025.
>
> [5] Vainer, Shimon, et al. "Collaborative control for geometry-conditioned pbr image generation." European Conference on Computer Vision. Springer, Cham, 2025.

---

> ### Author Response · Authors · 2024-12-02
>
> As the discussion period comes to a close, we wanted to kindly follow up on our latest response. We would greatly appreciate it if the reviewer could share any remaining concerns or insights before the discussion ends.
>
> If the response addresses the reviewer’s concerns, we would also kindly ask them to reconsider their score.
>
> We would like to thank the reviewer once again for their thoughtful feedback and time

---

### Official Review · Reviewer_Vjsj · 2024-11-03

**Soundness:** 3
**Presentation:** 3
**Contribution:** 3
**Rating:** 8
**Confidence:** 3

**Summary:**

This paper studies how to generate high quality mesh of objects with GIM.

The 3D objects are parameterized into 2D chart geometry image. This representation is compact where the prior of 2D SD model can be exploited and maintains good geometry and semantic properties. The generation happens vis modifying standard SD with cross attention between different attributes of the chart.

**Strengths:**

- The reviewer loves the elegant idea very much. I personally believe this is the correct way in a long-term to generate usable (industry level) and high-quality assets for objects. Because most generative model for objects now depends on large synthetic dataset, with known geometry.
- The system is simple and effective.
- Also, the semantically meaningful parameterization has much more space in the future for editing and manipulation.

**Weaknesses:**

- My main concern is maybe the diffusion overfit the proposed UV chart distribution. But we know that there is uncertainty of how an object can be parameterized into a chart. Is there any measurement of how good the chart is in the proposed preprocessing pipeline? e.g. it maintains the surface area or local differential properties? Can the proposed method generate different chart pattern for the exact same object? And how diverse is the chart?
- There are some small artifacts in the boundary of the mesh, which may due to the boundary of the chart. I believe this can be fixed in future works, but an analysis of this artifacts may provide more intuition of the future work.
- Maybe one or two applications like relighting .etc will future demonstrate the strength of generating object via GIM.

**Questions:**

Please see the first two questions in the weakness: the chart quality and diversity; and the artifacts on the boundary.

---

> ### Author Response · Authors · 2024-11-24
>
> We sincerely thank the reviewer for their kind words and insightful feedback. We are particularly grateful for the recognition of our approach's potential for generating industry-quality 3D assets in a long-term. Below, we address the specific questions and points raised.
>
> ### **Chart Quality and Diversity**
>
> To ensure high-quality parameterizations, we curated a subset of Objaverse [1] containing only handcrafted UV maps (we filtered out 3D scans and low-poly models, as well as UV maps with noisy boundaries and a high number of charts). These parameterizations were originally created for texturing purposes and adhere to principles such as minimizing distortion, optimizing seam placement, and often aligning charts with semantic parts of the object. While we did not explicitly measure these properties, we assume these parameterizations to be of high quality based on their intended use.
>
> Following [2], we rescaled individual charts based on their corresponding surface area (approximating an equal-area projection) to enhance uniformity across charts. Our preprocessing pipeline is fully deterministic, ensuring that the same layout is consistently produced for a given UV map. Initially, we experimented with randomized chart packing as an augmentation strategy, but we abandoned this approach due to the increased complexity it introduced.
>
> Our dataset spans a diverse object categories, ensuring a wide variety of underlying layouts. We believe that this diversity contributes to the robustness of our method and its ability to generalize across different objects and parameterizations (Figures 5-6).
> Additionally, we plan to update the paper with a study on layout novelty before the end of the discussion period.
>
> ---
>
> ### **Boundary Artifacts and Analysis**
>
> We have expanded the analysis of boundary artifacts and outlined potential solutions (see Section 4.5 and Appendix B).
>
> #### **Key Findings**:
> 1. **Lack of Inter-Chart Connectivity**: Multi-chart geometry images do not inherently encode connectivity information between adjacent charts, resulting in disconnected parts.
> 2. **Irregular Chart Boundaries**: These boundaries are prone to undersampling, leading to visible cracks, as analyzed in the original MGIM paper [3].
>
> The original MGIM paper proposed a zippering algorithm to address these issues. However, this approach assumes a watertight mesh, a condition not guaranteed in real-world datasets. Future work will focus on generalizing and adapting this algorithm. Additionally, sampling modifications and further VAE improvements could help mitigate these issues.
>
> Further analysis and visualizations of these artifacts are provided in Section 4.5.
>
> ---
>
> We greatly appreciate the reviewer’s feedback, which has been very helpful in refining our work.
>
> ---
>
> ### **References**
>
> [1] Deitke, Matt, et al. "Objaverse: A universe of annotated 3D objects." *Proceedings of the IEEE/CVF Conference on Computer Vision and Pattern Recognition*, 2023.
> [2] Sinha, Ayan, Jing Bai, and Karthik Ramani. "Deep learning 3D shape surfaces using geometry images." *European Conference on Computer Vision*, 2016.
> [3] Sander P, Wood Z, Gortler S, Snyder J, Hoppe H. "Multi-Chart Geometry Images." *Eurographics Symposium on Geometry Processing*, 2003.

---

> > ### Author Response · Authors · 2024-11-25
> > **UV layout novelty**
> >
> > We sincerely thank the reviewer again for raising this important concern. Below, we provide additional evidence to further clarify the model’s generalization capabilities.
> >
> > ### **UV layout novelty**
> >
> > To demonstrate our model's ability to generate novel UV layouts not present in the training data, we visualized the dataset entries closest to the generated samples in terms of layout similarity (see Figure 16). For this purpose, we constructed an index of binary multi-chart masks extracted from all geometry images in our dataset using the Faiss library [1]. We queried this index to find the nearest neighbors for the generated samples based on Euclidean distance.
> >
> > The visualization, provided in Appendix E, shows non-trivial differences between the generated layouts and those found in the training data. These differences highlight the model’s ability to generate novel layouts
> >
> > We hope this additional analysis addresses the reviewer’s concern regarding overfitting.
> >
> > ---
> >
> > ### **References**
> >
> > [1] Douze, Matthijs, et al. "The faiss library." *arXiv preprint arXiv:2401.08281* (2024).

---

> ### Comment · Reviewer_Vjsj · 2024-11-27
> **I keep my 8 rating**
>
> After reading other reviews and the author's response, I still believe this paper provides an interesting way (to be honest, I believe this GIM is a more correct way than many recent 3D generation representations, some recent young generations does not notice many classical ideas that should work elegantly).
>
> The MGIM boundary indeed creates some artifacts but I guess this can be fixed easily in the future. I personally weigh the novelty and the high-level thoughts more than the actual performance. Every paper has flaws but this paper I believe provides a novel and correct direction for 3D assets generation.
>
> I keep my original rating.

---

### Author Response · Authors · 2024-11-23
**Rebuttal**

We sincerely thank all reviewers for their thoughtful feedback and engaging questions.
We have revised the paper to address key concerns.


## Evaluation

The evaluation section has been significantly expanded, now including comparisons with recent methods such as MVDream[1], DreamGaussian[2], and CRM[3] (see Section 4.3). Visual comparisons have also been improved, with additional results in Figure 9. To provide a more comprehensive overview, we have included detailed information about our evaluation methodology, such as the full list of prompts and the specifics of adapting Image-to-3D models for Text-to-3D generation.

### Evaluation Results

| **Method**          | **Time**       | **UV-unwrapping**   | **Relightable** | **Separable** | **T³** ↑ | **AR > 4 (%)** ↓ | **RR > 4 (%)** ↓ | **MA < 10 (%)** ↓ |
|----------------------|----------------|---------------------|-----------------|---------------|----------|------------------|------------------|-------------------|
| **GaussianDreamer**  | 15 minutes     | No                  | No              | No            | 28.7    | -                | -                | -                 |
| **MVDream**          | 30 minutes     | XAtlas              | No              | No            | *47.8*  | -                | -                | -                 |
| **DreamGaussian**    | 7 minutes      | XAtlas              | No              | No            | 19.8    | -                | -                | -                 |
| **LGM**              | 1.8 minutes    | XAtlas              | No              | No            | 29.9    | 0.068            | 0.081            | 0.060             |
| **CRM**              | 13 sec         | XAtlas              | No              | No            | 39.6    | **0.012**        | **0.017**        | **0.013**         |
| **SF3D**             | **3.5 sec**    | Cube projection     | **Yes**         | No            | **48.2**| 0.112            | 0.117            | 0.122             |
| **Ours**             | *10 sec*       | **Generative**      | **Yes**         | **Yes**       | 35.2    | *0.038*          | *0.042*          | *0.035*           |

The results demonstrate that our method is competitive with state-of-the-art approaches on the T³ Bench[4], while also introducing novel capabilities. Notably, to the best of our knowledge, our approach is the first Text-to-3D method capable of natively generating UV maps directly from data (see Figure 10).


## Cracks and Other Limitations

We have also expanded the analysis of our method’s limitations, addressing the causes of cracks and proposing potential solutions and directions for future work (see Section 4.5, Appendix B). All revisions have been highlighted in blue for ease of reference.

In addition to these updates, we have prepared individual responses to each reviewer and welcome any follow-up discussions. Given the absence of major concerns and the consensus among reviewers that our paper is both novel and a meaningful contribution, we hope this rebuttal encourages reviewers to adjust their scores accordingly.



[1] Shi, Yichun, et al. "Mvdream: Multi-view diffusion for 3d generation." arXiv preprint arXiv:2308.16512 (2023).

[2] Tang, Jiaxiang, et al. "Dreamgaussian: Generative gaussian splatting for efficient 3d content creation." arXiv preprint arXiv:2309.16653 (2023).

[3] Wang, Zhengyi, et al. "Crm: Single image to 3d textured mesh with convolutional reconstruction model." European Conference on Computer Vision. Springer, Cham, 2025.

[4] He, Yuze, et al. "T $^ 3$ Bench: Benchmarking Current Progress in Text-to-3D Generation." arXiv preprint arXiv:2310.02977 (2023).

---

### Meta-Review · Area_Chair_QSuf · 2024-12-22

**Metareview:**

This paper introduces Geometry Image Diffusion, a novel framework for Text-to-3D generation that leverages 2D diffusion-based techniques to represent 3D shapes through geometry images. By sidestepping the complexity of existing 3D-aware architectures, the method significantly enhances efficiency and generalization in Text-to-3D tasks. The integration of a Collaborative Control mechanism further allows the model to exploit rich 2D priors from established Text-to-Image models like Stable Diffusion, resulting in high-quality and semantically meaningful 3D asset generation.

All reviewers provided positive ratings for this work, with several highlighting its novelty and potential impact on Text-to-3D generation. The authors also effectively addressed concerns raised during the review process, further strengthening the consensus around the paper’s merit. Based on the feedback and discussions, the area chair finds no reason to overturn the consensus and recommends acceptance. However, there are still areas that need further attention. Reviewer Vjsj noted that the MGIM boundary can produce some artifacts, and Reviewer nxjJ remains concerned about the diversity of the representation. The authors are encouraged to address these issues in the final version to further strengthen the paper.

**Additional Comments On Reviewer Discussion:**

The rebuttal effectively addressed the concerns. Reviewer Vjsj maintained a positive rating, while Reviewer nxjJ, Reviewer zGeZ, and Reviewer tGfn raised their ratings to borderline accept.

---

### Decision · Program_Chairs · 2025-01-22

Accept (Poster)